# Prevalence of Perineal Tear Peripartum after Two Antepartum Perineal Massage Techniques: A Non-Randomised Controlled Trial

**DOI:** 10.3390/jcm10214934

**Published:** 2021-10-25

**Authors:** María Álvarez-González, Raquel Leirós-Rodríguez, Lorena Álvarez-Barrio, Ana F. López-Rodríguez

**Affiliations:** 1Faculty of Health Sciences, Universidad de León, Astorga Ave. s/n, 24401 Ponferrada, Spain; dfimag01@estudiantes.unileon.es (M.Á.-G.); ana.lopez.rodriguez@unileon.es (A.F.L.-R.); 2SALBIS Research Group, Faculty of Health Sciences, Universidad de León, Astorga Ave. s/n, 24401 Ponferrada, Spain; rleir@unileon.es

**Keywords:** musculoskeletal manipulations, primary prevention, perineum, obstetric labour complications, physical therapy modalities, chronic pain

## Abstract

Perineal massage increases elasticity of myofascial perineal tissue and decreases the burning and perineal pain during labour, thus optimising child birth, although an application protocol has not been standardised yet. The objective of this study is to determine the efficiency of massage in perineal tear prevention and identification of possible differences in massage application. Total of 90 pregnant participants were divided into three groups: perineal massage and EPI-NO^®^ device group, applied by an expert physiotherapist, self-massage group, where women were instructed to apply perineal massage in domestic household, and a control group, which received ordinary obstetric attention. Results: The results showed significant differences among the control group and the two perineal massage groups in perineal postpartum pain. Correlations in perineal postpartum pain, labour duration and the baby’s weight were not statistically significant. Lithotomy posture was significantly less prevalent in the massage group than in the other two; this variable is known to have a direct effect on episiotomy incidence and could act as a causal covariate of the different incidence of episiotomy in the groups. Perineal massage reduces postpartum perineal pain, prevalence and severity of perineal tear during delivery.

## 1. Introduction

Perineal tear is a tissue lesion that occurs during vaginal delivery and is classified into four degrees: First-degree affects only the vaginal mucosa or perineal skin; second-degree affects perineum muscular fibres; in the third-degree the muscular lesion also affects anal sphincter muscles; and in fourth-degree tear the rectal mucosa is affected (resulting in communication of the vaginal epithelium and anal epithelium). Third-degree tears are subdivided into three types: (a) less than 50% of the external anal sphincter fibres are torn; (b) more than 50% of external anal sphincter fibres are torn; and (c) both external and internal anal sphincters are torn [1,2,3]. Third-and fourth-degree injuries are regarded as ‘high order’ since they are associated with higher risk of urogynaecological, sexual and colon-proctological dysfunctions than in first- and second-degrees [4,5]. Prevalence in developed countries varies, reaching 10.2% in women who had vaginal birth, although there is recognised lack of standardisation in perineal tear identification and is also under-diagnosed on numerous occasions [6,7]. All of these factors added to those that have an impact on women’s health (dyspareunia, chronic pelvic pain, vaginal flatulence, urinary, faecal or anal incontinence, prolapses…) make it difficult to collect data on the economic impact of perineal tears in the in participating health service system [8,9]. Risk factors of third- and fourth-degree tears are split into three subgroups: maternal (nulliparity, Asian ethnical origin, vaginal birth after caesarean section, less than 20 years of age and shortened perineal length), foetal (more than 4000 g of foetal weight, shoulder dystocia and occiput-posterior position) and intrapartum (use of medical equipment during delivery like forceps and vacuum, prolonged second stage of labour, epidural and oxytocin use, midline episiotomy and lithotomy or deep squatting position during delivery) [10,11,12,13].

Obstetric physiotherapy is a discipline that provides benefits to pregnant women [14,15]. It is mainly used with the following objectives: treatment of painful obstetric pathologies (lumbago, perineal pain, sciatica, cramps, etc.,) [16], physical preparation for delivery (perineal massage, pushing teaching, practice of different dilatation and second-stage labour positions, etc.,) [17,18] and to provide information on changes associated to delivery and postpartum (which psychologically prepares the patient for the upcoming changes and raises her self-confidence to deal with them) [19,20]. During the last two decades there has been a growing body of scientific evidence about the benefits and efficiency of perineal massage in prevention of perineal injuries during delivery and reduction of incidence and severity of perineal tear. The physiological explanation is that massage increases elasticity of myofascial perineal tissue and decreases the burning and perineal pain during labour, thus optimizing child birth [21,22], although an application protocol has not been standardised yet: manual therapy, its duration, its method of application (self-massage, applied by partner or physiotherapist), its frequency, start of application during pregnancy or suitability of instruments such as EPI-NO^®^ Childbirth Trainer (Tecsana, Munich, Germany) and/or simultaneous application of oils or lubricants with skin care substances [23,24,25].

The objective of the present investigation is to determine the efficiency of massage in perineal tear prevention and identification of possible differences in its mode of application (between self-massage and massage by a physiotherapist).

## 2. Materials and Methods

### 2.1. Experimental Design and Sample

A non-randomised controlled trial was made among women selected due to their interest in participating in the study through the information provided at their primary care attention centre (at their first obstetric consultation with midwife and/or gynaecologist or by information leaflets, handed at the care attention centre). Recruitment was carried out in three primary care centres depending on the same hospital, where all participants gave birth. During the year prior to this study (2019), 397 babies were born in the municipality where this study was carried out. Based on this data, the sample should have at least 59 pregnant women to reach 90% confidence level and a margin of error of 10%.

The following inclusion criteria were defined to participate in the study: (a) women between 18 and 40 years of age; (b) full term delivery (37th week or more); (c) single gestation and with cephalic presentation; (d) pregnancy with no complications, nor added risks during gestation; (e) no participation in any other psychoprophylaxis intervention; (f) deliver at the Hospital Nuestra Señora de Sonsoles (Spain). The following exclusion criteria were simultaneously defined: (a) Any contraindication for perineal massage and/or vaginal delivery; (b) medical diagnosis of any urogynecological pathology previous to gestation process; (c) any records of caesarean delivery and/or history of perineal injury; (d) not giving informed consent of participation in the study or lack of attendance to every programmed intervention and/or evaluation session.

None of the participants rejected nor abandoned the study. The sample consisted of 90 women (Figure 1).

### 2.2. Experimental Procedure

Participants were divided into three groups: Perineal massage and EPI-NO^®^ device group, applied by an expert physiotherapist (*n* = 30); self-massage group, where women were instructed in perineal massage application in domestic household (*n* = 30); and a control group (*n* = 30), which received ordinary obstetric attention (medical controls and regular information sessions by the midwife). Groups were not randomised since the priority was the wellbeing of the pregnant participant and were assigned according to the participant’s preferences (attendance at the medical centre for physiotherapy treatment, self-massage at home or attendance at childbirth preparation sessions only).

Approval for the study was obtained through the Ethics Committee of the University of Leon, Spain (code: ETICA-ULE-021-2018). All participants signed an informed consent form, in accordance with the Declaration of Helsinki (rev. 2013), and had the option to revoke their participation in the study at any time. Ethical regulations were respected as well as the Spanish Law for Protection Data Organic Law and for Biomedical Research in Human Participants.

Data collection took place during an evaluation session on the fifth- or sixth-postpartum week through a self-reported form where participants registered the characteristics of delivery (gestation week, baby’s weight, duration and posture of delivery, tear, episiotomy, use of equipment and/or analgesia). The form also included a question on intensity of perineal pain at the time of evaluation (quantified by visual analogue scale).

### 2.3. Perineal Self-Massage Intervention

Self-massage group received permanent instructions on perineal massage during pregnancy: it should be applied at least twice a week (on alternate days) during 10 min using a water-based lubricant from the 34th gestation week until delivery. The position during self-massage should be comfortable and relaxed (recommended: lying face up or semi-seated position). External massage should be applied in semicircles (towards the medial) on both sides of the vaginal vestibule and by ‘pumping’ (rhythmically pressing and releasing) on the central core of the perineum. Internal massage should be applied intracavitary with the thumb by dorso-central glides on both vaginal lateral walls (gliding on the introitus vaginae from 8 to 11 and from 4 to 1 clockwise until tissue relaxation). Then, with the help of the index finger, identify tenser areas on the vaginal walls (such as thin strings) that could be gently pressed or rubbed and then wait for relaxation. Finally, apply tissue stretching technique by clamp traction with one intracavitary finger and other external finger until tension or discomfort, hold the position until relief and then without releasing, apply traction again. This latter technique should be repeated three times on both sides (gliding on the introitus vaginae from 11 to 1 clockwise). Participants in the self-massage group received exhaustive information on massage application and filled in a daily register to check on their adherence to guidelines given and were also checked weekly (at consultation in person or by phone).

### 2.4. Perineal Massage Intervention

Perineal massage was applied by a physiotherapist expert in urogynaecology and obstetrics during a total of 6 to 10 sessions (from 34th gestation week until delivery) of 30 min each on a weekly basis. The position of the pregnant participant is semi-seated, with her back against the stretcher and the legs on the stretchers’ legs. A water-based lubricant was used for massage.

The treatment sequence included external massage with two manoeuvres: vulvar drainage and pump of the perineum central core. Drainage is a semicircular application with two or three fingers toward the medial on both sides of the vaginal vestibule, 15 semicircles on each side, in 3 series of 5 repetitions and moving forward in anteroposterior direction. In case of identification of vulvar oedema, there should be as many applications as needed. Pumps is a series of rhythmical pressures with one or two fingers (3 series of 5 repetitions, that is, 15 pumps in total). In case of muscular oedema or hypertone, as many pumps as needed should be applied until normalisation of tissue.

Intracavitary techniques were then applied through three manoeuvres. First, massage with longitudinal glides on the ani elevator muscles, on both lateral walls of the vagina in anteroposterior direction (following the track of the ischiopubian branches) with the second and third fingers. There were 3 series of 5 glides on both sides, increasing its number until normalisation of tissue. Second, application of treatment of myofascial trigger points of pelvic diaphragm identified by tense bands or active points that provoked referred pain recognisable to the patient and/or response to local spasm. In such cases, manual technique involves inhibition by pressure and/or friction until normalisation of tissue and disappearance of associated symptomatology. Finally, application of manual stretches on both sides of the vagina (on the area coinciding with a future possibility of lateromedial episiotomy). The stretching was applied in three stages, moving progressively, according to the sensations of the pregnant participant and the elasticity of the perineal muscles.

After application of manual techniques, instrumental massage was applied with the EPI-NO^®^ device. First, the device should be introduced deflated and once placed in the vagina, slowly inflated until finding the first barrier, where the pregnant participant manifests sensation of stretching (never discomforting nor painful). Inflation volume was held until the pregnant participant noted her sensation of stretching diminished, then the ball was slowly inflated again until meeting another stretching barrier. Three barriers should be met, and the device should be moved gently in the search for stretching barriers and local massage.

Finally, an external manual technique was applied to relax the perineal area through perineal global pumping by contacting with the first phalanx of the four last fingers of the hand (with fist closed) on the vulvar area. Total of 3 series of 5 pumps were applied.

### 2.5. Statistical Analysis

Statistical analysis was carried out by a blind researcher to experimental groups (unaware of the database codification of the three sample subgroups).

The sample was described through statistical descriptions (frequencies, percentages, average and standard deviation).

Kolmogorov-Smirnov tests and Levene’s test for equality of variances were applied to check the distribution of the data for the pre-treatment measure of the outcome variables in the three experimental conditions [26]. Since the results confirmed normal distribution and equality of variances, in categorical variables independent samples Chi-square test and Fisher exact test were used to verify the homogeneity of the groups, using Cramer’s V as measure of the effect sizes. Three groups repeated measures analysis of variance (ANOVAs) were used to assess the changes in clinical variables and psychosocial functioning, computing pairwise differences using Bonferroni correction, and partial eta-squared (η^2^p) was calculated to assess the effect sizes. All effect sizes were interpreted using the benchmarks provided by Cohen [27], (η^2^p: small < 0.06, medium > 0.06 and <0.14, and large > 0.14; Cramer’s V: small < 0.3; medium > 0.3 and <0.6, and large > 0.6).

A correlation analysis was made in perineal pain, duration of delivery and baby’s weight to find out the relationship between them. Furthermore, multinomial logistic regressions were conducted to identify the factors associated with less perineal tears (reference condition). Independent variables were inserted simultaneously into regression models for relative risk (RR) of each variable, which was controlled for all other covariates; the model was initially adjusted by age.

All statistical analyses were conducted with Stata v.12 (College Station, TX, USA) and statistical significance was established at *p* < 0.05 for all statistical tests.

## 3. Results

### 3.1. Descriptive Analysis

Descriptive analysis (Table 1) identified significant differences in age variable among the three subgroups (*p* < 0.05; η^2^p = 0.09) and for perineal postpartum pain variable between control group and the two perineal massage groups (*p* < 0.01; η^2^p = 0.1).

Characteristics of labour (Table 2) were statistically different only in relation to incidence of episiotomy (x^2^ = 20.47; *p* < 0.001; V = 0.48) and delivery posture (x^2^ = 14.66; *p* = 0.02; V = 0.29).

### 3.2. Bivariate Analysis

A correlation analysis was made in intensity of perineal postpartum pain, duration of delivery and the baby’s weight resulting both not statistically significant (*p* > 0.05).

On the other hand, intensity of postpartum perineal pain was significantly different in women with moderate or severe tear, compared to women with no tear (*p* = 0.009; η^2^p = 0.1) or mild tear (*p* = 0.004; η^2^p = 0.1), but no differences were found in the last two. ANOVA analysis between perineal pain and other obstetric variables in Table 2 was not significative.

### 3.3. Regression Analysis: Determinants in Perineal Tear

Table 3 describes the results of multinomial logistic regressions in dependent variables (massage group, posture of delivery, medical equipment used in child birth) and the result variable (tear). The later was significantly associated only to the massage group. Women who received massage were four times less likely to suffer from mild tear (RR = 0.25; *p* = 0.03) and 2.94 times less likely to suffer from moderate or severe tear (RR = 0.34; *p* = 0.003). On the other hand, self-massage only decreased the probability of suffering from serious tear by 1.12 times (RR = 0.89; *p* = 0.006).

## 4. Discussion

The objective of the present investigation is to determine the efficiency of massage in perineal tear prevention during labour and identify the differences (if any) between self-massage application and massage by a physiotherapist. According to the results obtained, massage is efficient in preventing perineal tears and there are differences between self-application and massage by a physiotherapist. Perineal tear prevalence of 40% was found in women of the control group, 30% in self-massage group and 26.6% in massage group.

The intensity of perineal pain was not associated with the baby’s weight or duration of delivery, episiotomy, posture of delivery or the use of equipment or analgesia during labour. This could be due to the fact that these variables were associated to perineal pain in immediate postpartum [28,29] since their influence could be decreased by the time of the evaluation of the present study. Perineal pain was actually significantly lesser in the group that received massage by a physiotherapist compared to the other two groups. Efficiency of perineal massage in prevention and treatment of postpartum perineal pain had been previously identified [21,22,30]. Nevertheless, it is the first time that the different effects of massage applied by a physiotherapist are compared to self-massage. Besides, the self-massage group reported less perineal pain that the control group, but not in a statistically significant way. This could be explained by the fact that the control group reported a low average pain intensity (again probably related to the time of evaluation of this variable).

Previous studies have identified the relation between perineal pain and perineal tear, which could be due to the musculoskeletal damage and sometimes nerve damage [29,31]. There were no differences identified in perineal pain between women with no tear and those with mild tear, which could lead to the conclusion that duration of pain caused by mild perineal tear does not last more than 6 weeks postpartum.

Statistically less incidence of episiotomy was identified in the massage group compared with the other two groups. There were no differences in episiotomy incidence between control and self-massage groups. Preventive effect of perineal massage on episiotomy was previously identified [30,32,33]. But it is the first time that the different efficiencies between perineal massage applied by a physiotherapist and self-applied massage are compared. Lithotomy posture was significantly less prevalent in the massage group than in the other two, this variable is known to have a direct effect on episiotomy incidence [34,35] and could act as a causal covariate of the different incidence of episiotomy identified in the groups. In contrast to previous studies [34,35,36], posture of labour did not reveal as predictive in tear incidence.

Similar to perineal pain analysis, tear incidence had significative results only for the treatment group as a predictive variable. The rest of the obstetric variables showed no predictive capacity on tears. Among the known effects of perineal massage is perineal tear prevention [22,37], and the analysis enabled the contrast of different effects between self-massage and massage applied by a physiotherapist.

The present study has methodological limitations that must be recognised such as the use of a small size sample, therefore generalisation of obtained data may be limited to the total population of pregnant women. On the other hand, it has not been possible to check the effect of the strange variable, that is, the posture of delivery in relation to the different incidence of tear among the groups (though the posture of delivery did not prove predictive in logistic regression analysis). Variables analysed in this study were registered by the participants in a self-reported form, therefore the information provided could be biased or unreliable. Finally, the duration of the second stage of labour (an important covariate in this research) could not be obtained and the lack of long-term follow-up of the participants limits the impact on clinical practice of the results obtained.

Simultaneously, this study has strengths that should be recognised, since it is the first study to contrast the efficiency between a home-based self-massage program and physiotherapy clinical treatment, with various results. Besides, multiple obstetric variables were found that could act as strange variables with a confirmed influence on the results obtained. Finally, it is important to note that this article details the procedures applied in the different sample groups. In the different interventions applied, the techniques applied, their sequencing and duration are detailed. Therefore, this manuscript also advances in the standardisation of the perineal massage procedure in its different modalities.

Therefore, the findings hereby presented should be considered by the obstetric health institutions and staff members responsible for gestation and deliveries so they implement and foster, when possible, physiotherapy treatment with perineal massage during pregnancy, and, when massage cannot be applied by a professional, sensitise pregnant women on the importance of self-massage as a self-care habit and preventive of obstetric perineal damage.

## 5. Conclusions

Perineal massage reduces postpartum perineal pain and prevalence and severity of perineal tear during delivery. Moreover, perineal massage applied by a healthcare professional has positive effects significantly better than those of self-massage.

Prophylaxis during pregnancy is of utmost importance to avoid obstetric perineal damage.

More investigations are needed on this field, especially when performing long-term follow-up studies, to be able to implement perineal massage through standardised criteria to protect the health of the mothers and that obstetric process causes the least possible damage.

## Figures and Tables

**Figure 1 jcm-10-04934-f001:**
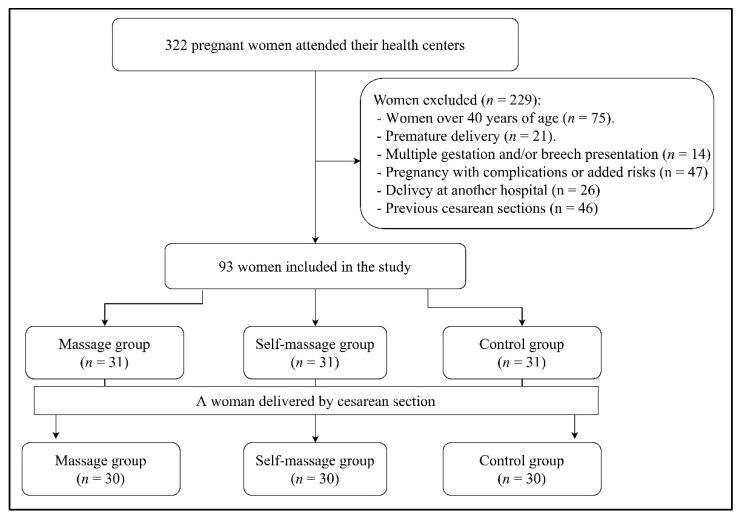
CONSORT flow diagram.

**Table 1 jcm-10-04934-t001:** Descriptive statistics of the sample (data provided: mean ± standard deviation).

	All (*n* = 90)	Control (*n* = 30)	Self-Massage (*n* = 30)	Massage (*n* = 30)
Age (years)	32.7 ± 3.9	31.4 ± 4.4 ^a,b^	33.2 ± 3 ^a^	33.6 ± 3.8 ^b^
Height (cm)	163.8 ± 6	163.2 ± 5.8	164 ± 5.2	164.2 ± 6.9
Weight (kg)	57.9 ± 7.9	58 ± 8.6	59 ± 8.5	56.7 ± 6.3
Body Mass Index (kg/m^2^)	21.6 ± 2.7	21.7 ± 2.6	21.9 ± 2.7	21.2 ± 2.8
Weight gain (kg)	12 ± 4.1	12.3 ± 4.5	12.4 ± 4.1	11.3 ± 3.7
Deliveries (n°)	1.4 ± 0.5	1.4 ± 0.5	1.3 ± 0.5	1.5 ± 0.6
Labour week (n°)	39.3 ± 1.8	38.8 ± 2.4	39.6 ± 1.5	39.4 ± 1.3
Baby weight (kg)	3.3 ± 0.4	3.2 ± 0.6	3.3 ± 0.3	3.3 ± 0.3
Duration of labour (hours)	10.7 ± 7.9	9.6 ± 6.6	12.8 ± 9.1	10 ± 7.9
Perineal pain (points)	2 ± 2.5	2.8 ± 3 ^b^	2.3 ± 2.5 ^c^	1 ± 1.5 ^b,c^

ANOVA significant results control vs. self-massage: ^a^ *p* < 0.05. ANOVA significant results control vs. massage: ^b^ *p* < 0.01. ANOVA significant results self-massage vs. massage ^c^ *p* < 0.01.

**Table 2 jcm-10-04934-t002:** Delivery characteristics [data provided: *n* (percentage)].

	All (*n* = 90)	Control (*n* = 30)	Self-Massage (*n* = 30)	Massage (*n* = 30)
Episiotomy	37 (41.1)	20 (66.7)	14 (46.7)	3 (10)
Perineal tear				
No	55 (61.1)	18 (60)	21 (70)	22 (73.4)
Mild	29 (32.2)	8 (26.7)	7 (23.3)	7 (23.3)
Moderate/severe	6 (6.7)	4 (13.3)	2 (6.7)	1 (3.3)
Position				
Lithotomy	69 (76.7)	27 (90.1)	24 (80)	18 (60.1)
Sideways	5 (5.6)	1 (3.3)	3 (10)	1 (3.3)
Sit/squat	13 (14.4)	1 (3.3)	2 (6.7)	10 (33.3)
Standing	3 (3.3)	1 (3.3)	1 (3.3)	1 (3.3)
Instrumental				
No	72 (80)	20 (66.7)	25 (83.3)	27 (90)
Vacuum	11 (12.2)	6 (20)	2 (6.7)	3 (10)
Forceps	7 (7.8)	4 (13.3)	3 (10)	0 (0)
Analgesia				
No	19 (21.1)	6 (20)	5 (16.7)	8 (26.7)
Local	2 (2.2)	1 (3.3)	1 (3.3)	0 (0)
Epidural	69 (76.7)	23 (76.7)	24 (80)	22 (73.3)

**Table 3 jcm-10-04934-t003:** Multinomial logistic regression of perineal tear in relation to massage group, weight gain, delivery position, analgesia and instrumental applied, adjusted by age.

Variable	Mild Tear		Moderate/Severe Tear	
	RR	95% CI	RR	95% CI
Massage group				
Control	1		1	
Self-massage	2.22	0.68–7.25	0.89 **	0.68–9.63
Massage	0.25 *	0.02–10.32	0.34 **	0.13–7.82
Weight gain				
<10	1		1	
11–15 kg	1.78	0.67–4.73	0.35	0.02–3.71
16–20 kg	0.95	0.21–4.27	0.88	0.08–8.98
21–25 kg	1.27	0.1–15.5	2.13	0.02–9.36
Delivery position				
Lithotomy	1		1	
Sideways	1.65	0.25–10.67	3.98	2.36–9.36
Sit/squat	2.12	0.63–7.14	4.28	0.69–12.37
Standing	4.95	4.23–57.9	5.23	4.63–11.68
Analgesia				
No	1		1	
Local	1.11	0.06–20.48	3.75	1.69–8.31
Epidural	0.45	0.16–1.28	1.06	0.11–10.12
Instrumental				
No	1		1	
Vacuum	0.38	0.07–1.88	4.4	0.64–30.4
Forceps	0.34	0.08–1.88	1.88	0.45–16.35
Episiotomy				
No	1		1	
Yes	0.4	0.06–1.96	1.4	0.02–13.54
Massage group	0.48 *	0.01–3.14	0.53 **	0.26–8.35
Weight gain	1.11	0.64–1.91	0.67	0.2–2.26
Delivery position	1.54	0.93–2.53	2.2	0.36–6.32
Analgesia	0.67	0.4–1.12	1.05	0.33–3.31
Instrumental	0.56	0.25–1.25	0.88	0.26–3
Episiotomy	0.49	0.36–1.68	0.59	0.26–2.69

The base outcome is “no perineal tear”. RR: relative risk; 95% CI: 95% confidence interval. * *p* < 0.05; ** *p* < 0.01.

## Data Availability

The data presented in this study are available on request from the corresponding author.

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
