# Peer review of "Prevalence of Perineal Tear Peripartum after Two Antepartum Perineal Massage Techniques: A Non-Randomised Controlled Trial"

_jcm, 2021, doi:10.3390/jcm10214934_

Round 1
Reviewer 1 Report
The study describes the anterpartum perineal massage which is increases elasticity of myofascial perineal tissue and decreases the perineal tear or pain during labour.
The objective of this study is to determine the efficiency of massage in perineal tear prevention and identification of possible differences in massage application.
It would be much appreciated if the Authors could be able to describe some recommendations for standardized criteria of procedures.
The corresponding links to Kolmogorov-Smirnov tests and Levene’s test should be cited in the References.
Author Response
Dear Editor and Reviewer of Journal of Clinical Medicine:
Thank you very much for your suggestions and contributions to improve the quality of the manuscript. Following your indications, we respond, point by point, to the reviewers' comments.
In the text, all the modified or added sentences have been written in red to facilitate the correction by the reviewers.
- It would be much appreciated if the Authors could be able to describe some recommendations for standardized criteria of procedures.
The authors have added this aspect at the end of the Discussion section.
However, we have doubts whether we have interpreted their correction correctly. If not, please let us know and we will correct the manuscript according to your recommendations.
- The corresponding links to Kolmogorov-Smirnov tests and Levene’s test should be cited in the References.
The authors have added a bibliographic reference that supports the use of these two statistical techniques.
Once again, thank you very much for the time spent and the interest shown in this work; as well as in the positive evaluations you have given of it.
Receive a warm greeting,
The authors.
Reviewer 2 Report
You should continue, mainly to increase the number of participants. Probably you should find proper way to randomise participants.
Author Response
Dear Editor and Reviewer of Journal of Clinical Medicine:
Thank you very much for your suggestions and contributions to improve the quality of the manuscript. Following your indications, we respond, point by point, to the reviewers' comments.
In the text, all the modified or added sentences have been written in red to facilitate the correction by the reviewers.
- You should continue, mainly to increase the number of participants. Probably you should find proper way to randomise participants.
The authors are grateful for the reviewer's suggestions for improvement and are aware of the limitations of our research. We will improve the limitations of this study in our next research.
In addition, we have modified the limitations section (at the end of the Discussion) to explicitly reflect the methodological limitations of this study.
Once again, thank you very much for the time spent and the interest shown in this work; as well as in the positive evaluations you have given of it.
Receive a warm greeting,
The authors.